# Severity Classification of Laboratory Animal Procedures in Two Belgian Academic Institutions

**DOI:** 10.3390/ani13162581

**Published:** 2023-08-10

**Authors:** Stéphanie De Vleeschauwer, Kathleen Lambaerts, Sophie Hernot, Karlijn Debusschere

**Affiliations:** 1Laboratory Animal Center, KU Leuven, 3000 Leuven, Belgium; 2Laboratory for In Vivo Cellular and Molecular Imaging (ICMI-MIMA/BEFY), Vrije Universiteit Brussel, 1090 Brussels, Belgium; sophie.hernot@vub.be; 3Core Facility ANIM, Vrije Universiteit Brussel, 1090 Brussels, Belgium; karlijn.debusschere@ugent.be; 4Core ARTH, Animal Facility, Ghent University, 9000 Ghent, Belgium

**Keywords:** severity classification, animal procedures, EU Directive, animal ethics committee, animal welfare body

## Abstract

**Simple Summary:**

According to European regulations, the severity of the suffering of animals during animal experiments should be assessed. Regulatory documents and guidelines provide recommendations on how to approach this severity assessment; however, they are often not specific enough, resulting in inconsistencies between different institutes performing the same procedures. To overcome this, two Belgian academic institutions with a focus on biomedical research, collaborated to develop and align the severity classification for all procedures performed. This was performed based on the available literature and guidelines, as well as the professional judgment of the designated veterinarians, animal welfare bodies and animal ethics committees. Throughout the manuscript, we motivate which criteria were used to classify procedures or groups of procedures within a specific category. Our collaborative classification includes many procedures and disease models in a variety of animal species for which a severity classification was not reported so far, or the terms that assign them to a different severity were too vague. We believe this extensive list of procedures and the approach described in this paper could be of great value to other research institutions.

**Abstract:**

According to the EU Directive 2010/63, all animal procedures must be classified as non-recovery, mild, moderate or severe. Several examples are included in the Directive to help in severity classification. Since the implementation of the Directive, different publications and guidelines have been disseminated on the topic. However, due to the large variety of disease models and animal procedures carried out in many different animal species, guidance on the severity classification of specific procedures or models is often lacking or not specific enough. The latter is especially the case in disease models where the level of pain, suffering, distress and lasting harm depends on the duration of the study (for progressive disease models) or the dosage given (for infectious or chemically induced disease models). This, in turn, may lead to inconsistencies in severity classification between countries, within countries and even within institutions. To overcome this, two Belgian academic institutions with a focus on biomedical research collaborated to develop a severity classification for all the procedures performed. This work started with listing all in-house procedures and assigning them to 16 (sub)categories. First, we determined which parameters, such as clinical signs, dosage or duration, were crucial for severity classification within a specific (sub)category. Next, a severity classification was assigned to the different procedures, which was based on professional judgment by the designated veterinarians, members of the animal welfare body (AWB) and institutional animal ethics committee (AEC), integrating the available literature and guidelines. During the classification process, the use of vague terminology, such as ‘minor impact’, was avoided as much as possible. Instead, well-defined cut-offs between severity levels were used. Furthermore, we sought to define common denominators to group procedures and to be able to classify new procedures more easily. Although the primary aim is to address prospective severity, this can also be used to assess actual severity. In summary, we developed a severity classification for all procedures performed in two academic, biomedical institutions. These include many procedures and disease models in a variety of animal species for which a severity classification was not reported so far, or the terms that assign them to a different severity were too vague.

## 1. Introduction

According to EU Directive 2010/63, a ‘procedure’ means any use, invasive or non-invasive, of an animal for experimental or other scientific purposes, with known or unknown outcomes, or educational purposes, which may cause the animal a level of pain, suffering, distress or lasting harm equivalent to, or higher than, what could be caused by the introduction of a needle in accordance with good veterinary practice. It includes any course of action intended, or liable, to result in the birth or hatching of an animal or the creation and maintenance of a genetically modified animal line in any such condition but excludes the killing of animals solely for the use of their organs or tissue. The Directive also states that all procedures on animals should be classified as ‘non-recovery’, ‘mild’, ‘moderate’ or ‘severe’. Severity classification of animal procedures does not only create opportunities to refine procedures, but it is also crucial in the harm–benefit analysis performed during the project’s evaluation and authorization [1,2]. The severity of procedures should be classified prospectively and retrospectively. The prospective severity classification is performed upfront during project writing and will thus help in the harm–benefit analysis. The retrospective, or actual severity, is estimated for each individual animal and is based on that animal’s experience during the course of the procedure. The latter enters the annual statistics that promote transparency and dictate public opinion.

However, the estimation of the level of pain, suffering, distress and lasting harm, and thus classifying the severity of a procedure, is not easy. Annex VIII of the EU Directive gives some examples of the severity classification of procedures. Although some classifications are very straight forward, e.g., categorizing ‘non-invasive imaging of animals with appropriate sedation or anesthesia’ as mild, other examples remain vague with room for interpretation. For example, ‘breeding of genetically altered (GA) animals, which is expected to result in a phenotype with mild effects’ does not provide clear guidance on what these mild effects are. Since the implementation of the Directive by the different EU member states, several articles have been published on the topic [3,4,5]. In addition, different working groups (Working Group of Berlin Animal Welfare Officers [6], EU severity assessment framework [7]) and countries (Switzerland [8] and UK [9]) have written guidelines on the severity classification of animal procedures. However, due to the large variety of disease models and animal procedures carried out in multiple animal species, guidance on the severity classification of specific procedures or models is often lacking or not specific enough.

Furthermore, the reported severity classifications per country within the EU depict large differences in the percentage of mild, moderate and severe procedures (Table 1).

These differences may be due to different research areas but also due to different severity classifications by the competent authorities. Within the EU, Belgium is one of the few countries where the competent authority has delegated project evaluation and authorization, and thus severity classification, to the institutional level instead of the regional or national level. As a result, the differences in severity classification for the same procedure probably occur even between different institutions within the same country.

Here, we want to share the prospective severity classification we have developed and aligned for two Belgian academic institutions. Severity classification was based on existing guidelines and the professional judgment of all parties involved. This was performed for all procedures performed in (laboratory) animals in two Belgian academic institutions with a focus on biomedical research.

## 2. Materials and Methods

### 2.1. Establishing a Framework for Severity Classification

In 2016, procedures and disease models performed at KU Leuven were listed by reviewing all ethical projects approved in the period 2013–2015. Although this catalog was not made within the scope of this project, it formed the basis for our severity classification. In 2019, the Norwegian Consensus-Platform for the Replacement, Reduction and Refinement of animal experiments (Norecopa) compiled available severity classifications from the EU Directive Annex VIII [7], the FELASA/ECLAM/ESLAV report [3], the UK Home Office [9], the Swiss FSVO (Federal Food Safety and Veterinary Office) [8] and the Working Group of Berlin Animal Welfare Officers [6]. At that time, KU Leuven decided to develop an in-house severity classification system for all procedures and disease models within KU Leuven. This work started by reviewing the literature and the previously cited legislation and guidelines that were consulted via the Norecopa database [11]. By doing this, it became clear that not all procedures from KU Leuven were listed and that some procedures were classified differently by the different guidelines. For example, gavage is classified as mild in the EU Directive while the Swiss FSVO considers this as moderate; a single subcutaneous (SC) injection is considered mild according to UK guidelines, while the Swiss FSVO considers this below threshold. Therefore, we decided to develop a more detailed severity classification at KU Leuven. During the process, another academic institution (VUB) was approached as we wanted to align the severity classification system with another Belgian academic institution. Together, we defined the following key points for our severity classification system:-It must be clear with defined cut-offs, avoiding (where possible) vague terms;-It should not only be a list of procedures within a severity classification but must also define common denominators so that it can easily be adapted when new models are adopted in an institution;-It must represent the severity of a procedure that is performed once by a well-trained person using the most refined technique. It must be applicable to all species and procedures.

### 2.2. Approach to an Institutional Severity Classification System

The development of the severity classification system is based on existing guidelines and professional judgment and is performed in a stepwise approach. First, all procedures from the catalog of procedures were divided into categories. Next, for each (sub)category, the aforementioned references and guidelines were extensively reviewed. Based on this, we decided which format of the available criteria and guidelines could be useful to approach the severity classification (see results). As mentioned above, guidelines were inconsistent for some procedures or missing for other procedures. When differences occurred between existing guidelines, the experts decided on the appropriate severity classification. When procedures were not reported in existing guidelines, the experts first defined criteria and common denominators to classify the severity of these (see results). If necessary (i.e., when experience with or knowledge of certain procedures or models was insufficient), research experts were consulted. This was mainly performed for the behavioral tests, pain tests and infectious disease models. Next, per (sub)category, an initial draft was drawn containing the severity classification proposal, references from other guidelines and, if necessary, references from the literature describing, e.g., clinical signs in certain disease models. The content of this initial draft was discussed until consensus was reached. If needed, additional references and (research) experts were consulted. Notably, consensus was usually reached easily. Once the initial draft was finalized, a final draft was sent to the whole institutional AEC (and AWB for VUB) for approval. AECs only evaluated procedures performed in their own institution. Once approved, the severity classification was adopted in the institutions.

### 2.3. Experts Involved

The following experts were involved in drafting and evaluating the severity classification:-The designated veterinarians from both institutions. They both have been working as designated veterinarians in academic institutions for more than 5 years. As members of different institutional AECs, they are closely involved in severity classification of procedures and research projects;-Subgroup of the KU Leuven AEC: to facilitate the process, a subgroup of the EC was appointed for this work. This subgroup consisted of the designated veterinarian, the animal welfare officer of the aquatic species and one of the external members of the AEC, a veterinarian who has been working as a designated veterinarian and consultant in laboratory animals for many years;-AWB: at VUB there is one central AWB in contrast to KU Leuven where each department has its own AWB. This means that the involvement of the AWB of the KU Leuven, in contrast to the VUB, is limited to some members of the AWB that were represented in other categories of experts;-AECs of both institutions. At VUB, the AEC consists of 32 members (members + substitutes), at KU Leuven of 27 members (members + substitutes);-Research experts: Researchers that were contacted to obtain more information on disease models were considered experts when they worked with a specific animal model for many years and had multiple peer-reviewed papers describing and utilizing a specific animal model.

The involvement of the different experts in the different steps of the process is indicated in Table 2.

## 3. Results

In the period of 2013–2015, a total of 804 projects involving animals were approved by the institutional AEC of KU Leuven. From these projects, 673 ‘technical’ procedures in 14 species were identified. These species were as follows: mouse, rat, rabbit, pig, sheep, rhesus macaque, zebrafish, killifish, chicken, xenopus, gerbil, guinea pig, hamster and calf. The technical procedures were divided into the following (sub)categories: administration, sampling, surgery and surgical induction of disease, behavioral testing, pain tests, imaging and function measurements. To cover the disease models, the following categories were added: chemical disease, wherein a disease is induced by the administration of a chemical, neoplasm, infectious disease and ‘disease models—others’, for those models not qualifying the aforementioned subcategories. GA lines with a harmful phenotype, abnormal housing and nutrition and clinical signs formed separate categories. These 16 subcategories were assigned to the following four larger categories entitled: procedures, measurements and tests, disease models and animals (Figure 1). As in vivo pharmacokinetics/pharmacodynamics and toxicity tests are not often performed in both institutions, no separate category was made for these tests in contrast to most available guidelines. Where necessary, the animal species are specified. If species is not specified, the severity level is considered to be the same across species, although not all procedures are carried out in all species.

As mentioned before, the factors that were thought to be crucial for the severity classification of a specific (sub-)category were determined, such as clinical signs, dosage or duration. Based on these factors, defined cut-offs were established, and procedures were assigned to different severity levels. The factors used and/or the final severity classification are not always according to the existing guidelines. This is especially true in the disease models, as they often depend on different parameters and can be progressive. Therefore, clinical signs were used as a means to classify the severity of the different disease models.

In the following chapters, we will describe in more detail which factors were used to assign a severity classification to a procedure. As some subcategories are classified similarly, the classification is discussed in the same section of the text. Appendix A provide an overview of all procedures with their severity classification. We have underlined the procedures/models for which severity has not been described elsewhere.

### 3.1. Severity Classification of Administration

To determine the severity of compound administration, we focused on the technical procedures required to administer a compound and not on the effects of the compound given. For these compound-specific effects, we refer to the chapters and Appendix A on disease models (chemical, infectious, neoplasm and others).

Following the EU Directive and UK guidelines, we classified all routes of administration in all species as mild. This includes conventional routes, like intravenous (IV) and SC injections, but also less conventional routes, such as intragastric administration in mouse pups. Some routes, such as intranasal administration, are considered mild as long as they are performed under anesthesia. Hydrodynamic tail vein injection (HDTVI) and injection in the footpad were classified as moderate as HDTVI can lead to transient, but severe, cardiovascular effects [12,13], and injections in the footpad are painful due to swelling in weight-bearing structures. Interestingly, our classification diverged most from the Swiss guidelines, in which some administration methods, such as a single SC or IV injection, are considered below threshold and gavage is considered moderate.

Appendix A describes the different severity classifications of administration.

### 3.2. Severity Classification of Sampling

All technical procedures involving taking fluid or tissue samples, including sampling for genotyping, were grouped into the category ‘sampling.’ Fluid sampling includes all fluids that can be sampled in different species, such as blood, urine and cerebrospinal fluid. For blood sampling, the withdrawn volume, whether it is replaced or not, and the technique used, were taken into account to assign the severity level. However, we did not determine a severity classification for serial blood sampling, as the frequency, time interval between samples and volume taken can all influence the final severity and, as such, need to be evaluated on a case-by-case basis. According to the Commission Implementing Decision (EU) 2020/569 of 16 April 2020 [14] and the EU Framework for the genetically altered animals [15], tissue sampling for genotyping is not considered a procedure if the sample obtained is a by-product from identification. However, if it is not a by-product from identification, it is considered a procedure and has thus been classified accordingly. Furthermore, ‘oocyte collection in xenopus by gentle squeezing’ and ‘non-invasive mucus sampling for genotyping zebrafish’ were included, both technical procedures for which a severity classification was not reported so far. Overall, most techniques are classified according to EU Directive and legislation.

Appendix A describes the different severity classifications of fluid and tissue sampling.

### 3.3. Severity Classification of Anesthesia, Surgery and Surgical Induction of Disease

Following EU Directive and UK guidelines, anesthesia as such is prospectively classified as mild.

To classify the surgical procedures, including those used to induce a specific disease model, a differentiation between minor and major surgery was made. Although the definition of minor and major surgery is still under debate [16,17], we defined minor surgery as surgery not opening body cavities and major surgery as surgery opening body cavities, such as the abdomen or thorax. Most types of minor surgery are classified as mild, and most types of major surgery, with appropriate analgesia, as moderate. However, the consequences of the (minor or major) surgical procedures should also be taken into account. Therefore, we also considered the following: impairment of locomotion, loss of function, rejection of organ transplanted and failure of the device implanted. As a result, we have stratified the severity of cardiac assist device implantation, stroke and myocardial infarction into two different severity categories, i.e., moderate and severe, rather than assigning them to a single severity classification as the examples provided by the EU Directive and the Swiss guidelines. In the case of cardiac device implantation, the presence or absence of a functional heart determines if the procedure is moderate or severe, respectively. In the case of myocardial infarction and stroke, the size, and, thus, effects on the animal determine whether this procedure is classified as moderate or severe.

Appendix A describes the different severity classifications of surgery and surgical induction of disease.

### 3.4. Severity Classification of Clinical Signs

Disease models are challenging to classify, as often diseases are progressive, i.e., worsening over time. Depending on the study objective, researchers may be interested in the early or late stages of a disease. To classify the severity of disease models, we therefore, decided to estimate the severity of disease models mainly based on the severity of clinical signs. Hence, much effort was put into the severity classification of clinical signs. Not only do these enable us to classify disease models we currently have at our institutions, but they will also facilitate severity assessment in case new models are developed. Furthermore, they also aid in the assessment of actual severity.

Clinical signs and their severity classification were developed for mammals, zebrafish larvae (up to 12 days post fertilization) and sexually mature zebrafish. We have defined clear cut-offs between severity levels, taking into consideration different parameters, such as duration, effect on behavior, etc. For the body weight loss of mammals, we took into consideration the developmental stage and timeframe wherein the animal loses weight. This is in line with UK guidelines. The severity classification of the other clinical signs of mammals is largely consistent with the available guidelines, although we included duration for some clinical signs. As at our institute, we also have disease models involving zebrafish larvae (independently feeding and thus covered by Directive 2010/63/EU); we established parameters for clinical follow-up of zebrafish larvae up to 12 days post fertilization. The following clinical signs are described and assigned to mild, moderate or severe: overall morphology, necrosis, swim bladder, posture, cardiac function and touch response. To our knowledge, clinical signs and severity classification for zebrafish larvae have not yet been reported. At our institute, there are currently no disease models involving adult zebrafish. Nonetheless, clinical signs and severity for adult zebrafish were included in the table. Therefore, we mainly followed Sabrautzki et al. [18], who defined and assigned scores to different clinical signs in adult zebrafish.

Appendix A describes the different severity classifications of clinical signs.

### 3.5. Severity Classification of Disease Models

As mentioned above, we have defined disease models as follows: chemical, infectious, neoplasm or others. As we do not have a separate category for toxicity tests, we included these in the chemically induced models. Although very different in etiology, they all result in the animals developing general or organ-specific clinical signs. Although the clinical signs seen in a disease model are usually the same, the severity of these signs may depend on several factors.

In the case of chemically induced disease models, the severity of the clinical signs may depend on the dosage and the type of chemical given. As this may differ depending on the study objective, we chose to prospectively classify this type of model according to the severity of the clinical signs expected using a specific dosage of a compound rather than assigning a fixed severity level. For example, intestinal inflammation in dextrane sulfate sodium (DSS) colitis in mice depends on many factors, such as DSS molecular weight and dosage, mouse strain, etc. [19,20]. Depending on the study objective, mice may experience mild to severe clinical signs. The severity classification should therefore be based on the severity of these clinical signs. The same is true for, e.g., diabetes. For well-established chemically induced models that always give similar clinical signs, the severity level is fixed. For instance, models of LPS-induced acute respiratory distress are always classified as severe. To classify the severity of toxicity tests at our institution, mainly conducted on zebrafish larvae, we took the following factors into consideration: clinical signs and, as described by Hawkins et al. [5], predictability and death as a possible outcome.

In infectious diseases, the clinical signs depend on the pathogen strain and dosage, the route of administration and animal species and the strain, making severity classification even more challenging. For the standardized models (using a specific pathogen and animal strain, dosage and route of administration), the clinical signs are known, and a specific prospective severity level is assigned. For example, Zika, dengue and Japanese encephalitis in AG129 mice always lead to severe clinical signs and are thus classified accordingly. Some infectious disease models are not so standardized, for instance, models of new, emerging pathogen strains. Prospective severity classification for these models is difficult, and we have therefore chosen not to include these.

For severity classification of (mouse) cancer models, we follow the same reasoning, i.e., assign severity level based on the clinical signs expected. As cancer is a progressive disease, these clinical signs will depend on study duration, objective and humane endpoints applied. In contrast to other disease models, we here refer to a specific set of clinical signs that can be expected in most cancer models. 

The disease models categorized in ‘others’, are those that do not fit into one of the above categories as they are induced by laser, hypoxia or hyperoxia, mechanically or by irradiation. Again, in this subcategory, severity is classified for different models that were not described before.

Appendix A describes the different severity classifications of chemically induced disease models.

Appendix A describes the different severity classifications of infectious disease models.

Appendix A describes the different severity classifications of neoplasm.

Appendix A describes the different severity classifications of other disease models.

### 3.6. Severity Classification of Abnormal Housing and Nutrition

Some procedures require changes in housing and/or nutrition. For alterations in both housing and nutrition, we mainly followed the Swiss FSVO guidelines, taking into consideration the duration of the abnormal housing and several other factors, such as social isolation. For food deprivation, the severity is based on body weight loss. For water restriction, a differentiation is made between feeding dry food and food containing fluids with different cut-offs in time. In the latter case, food restriction is only classified as severe if it is accompanied by dehydration. In general, instead of using non-specific terms, such as ‘a short period of time’, we rather specify the duration of the abnormal housing/nutrition, linking this to the appropriate severity. In our classification, no specific examples per species are given to make the use of these tables as broad as possible. As mentioned before, this does not necessarily mean that all these abnormal housing and nutrition conditions are used in all species.

Appendix A describes the different severity classifications of abnormal housing and nutrition.

### 3.7. Severity Classification of Behavioral Testing, Function Measurements and Imaging

There is a plethora of behavioral tests and function measurements. The difference between both terminologies is not always clear nor well described. We have therefore defined behavioral tests as all tests measuring fear, cognition, memory, etc. Function measurements, on the other hand, were defined as all tests measuring a body function, including motor function.

Starting from an overview of both behavioral tests and function measurements performed at our institutions, we searched for common denominators to classify them into different severity categories. The criteria taken into account are the following: handling and change in the environment during the test, changes in housing and nutrition (see above), invasiveness and the duration of restraint in case the procedure is being performed in an awake animal.

All imaging techniques, including those requiring injection of tracers or contrast, are classified as mild when performed under anesthesia. In the case no anesthesia is used and restraint of the animal is thus required, the severity is determined by the duration of restraint. A cut-off of one hour was determined to shift from mild to moderate severity.

Appendix A describes the different severity classifications of behavioral testing.

Appendix A describes the different severity classifications of function measurements.

### 3.8. Severity Classification of Pain Tests

For the pain tests, we took the duration and intensity of pain caused into account. By doing this, we mainly followed the Swiss FSVO guidelines and added just a few changes. Based on the retrospective severity assessment, we decided to classify nerve crush and ligation as moderate and not severe. Furthermore, writhing is always considered to cause severe pain [21]. For the footpad injections, we considered the effects caused by the injected compound. This results in a severe classification for the footpad injection of Complete Freund’s Adjuvant, capsaicin and acrolein and a moderate severity for footpad injection of saline and pregnolone sulfate.

Appendix A describes the different severity classifications of pain tests.

### 3.9. Severity Classification of GA Lines with Harmful Phenotype

As stated in the EU Directive, GA animals with a harmful phenotype must be classified as mild, moderate or severe. The determination of the severity of GA animals remains, however, very difficult. Defining what is considered harmful and what is not is the first challenge. Alterations may result in phenotypes that are macroscopically visible but do not necessarily affect the welfare or wellbeing of the animals. Therefore, we choose to classify all genetic alterations causing macroscopic changes not affecting welfare, as well as all phenotypic changes that can only be detected using specific testing (e.g., by behavioral tests or blood analysis), as not harmful. Another difficulty in assessing the severity of GA lines is that, in contrast to other categories, the assessment should be performed on the line and not the individual animal. Consequently, in the severity assessment, we consider the life-long effects of the GA, not taking into account the humane endpoints applied. Naturally, this does not mean that humane endpoints should not be applied. As a result, we classify some lines, especially those with progressive diseases such as cancer, as severe instead of stratifying the severity based on humane endpoints applied as Zintsch et al. [4] or the Swiss guidelines [8]. Here again, the classification is mainly based on clinical signs seen during the assessment of the line. Of note, our institutions only have GA rodent and zebrafish lines.

Appendix A provides an overview of the severity classification of GA lines with harmful phenotypes.

### 3.10. Severity Classification of Fetuses and Premature Animals

Article 2 of the EU Directive states that the Directive should apply to live, non-human vertebrate animals, including the following: independently feeding larval forms and the fetal forms of mammals from the last third of their normal development. Consequently, the fetuses of mammals in the first and second semester of gestation, birds before hatching and non-independently feeding larval forms from aquatic species are not considered experimental animals. Therefore, their use is not regulated unless procedures carried out could result in pain, suffering, distress or lasting harm if the fetuses are allowed to live beyond the first two-thirds of their development. According to Commission Implementing Decision (EU) 2020/569 of 16 April 2020 [14], fetal and embryonic forms of mammalian species shall be excluded from the provision of annual statistical data. Only animals that are born, including by cesarean section, and live are to be counted. To our knowledge, this means that for procedures on mammalian fetuses and early developmental stages in other species, severity should only be assessed for those animals that are born and in which pain, suffering or distress occurs or is likely. We report here the severity of two such models for which the severity was not reported before, i.e., in utero creation of spina bifida in lambs and growth retardation in rabbits.

Appendix A provides an overview of the severity classification of procedures carried out in early developmental stages.

## 4. Discussion

This manuscript describes the severity classification of laboratory animal procedures performed in two Belgian academic institutions with a focus on biomedical research. We explain how the process of severity classification is performed, and, more importantly, we describe the severity classifications for procedures and disease models through the tables presented in the Appendix A. Our goal was to be as specific as possible and provide clear cut-offs between categories. In addition, we included many disease models for which severity classification was not yet described. Those novelties are made clear by underlining them in the Appendix A.

Although performed with great care, this study has some shortcomings. Evidence-based severity classification is upcoming and has been performed for different procedures and models, such as epilepsy [22,23,24], repeated anesthesia [25,26], depression [27] and models of gastrointestinal diseases [28]. Recently, a mathematical model to estimate the severity of animal procedures has been described by Morton [29]. These methods use objective parameters, scores and tests to assess the severity of animal procedures. The severity classification described in this manuscript is based on available guidelines, in-house expertise as well as retrospective analyses, and although we are aligned with other guidelines, it is inevitably partly subjective. Further evidence-based severity classification of the different animal procedures is necessary, especially since the severity of procedures is an essential part of the harm–benefit analysis performed during project evaluation. It should thus give a correct reflection of the expected suffering of an animal during a certain procedure.

As we have covered all species used at our institutions and mostly used a one-fits-all approach (i.e., not distinguishing between species), species-specific behavior and sensory elements might not have been highlighted sufficiently. For example, we do not distinguish between the different (social) species when classifying abnormal housing and nutrition. However, the well-being of an individual animal might be affected differently depending on the species and sometimes even the sex. Indeed, some studies show no difference in the behavior between single-housed versus group-housed male mice [30], while others consider the single housing of male rats as severe [31]. These differences might be even more pronounced in non-rodent species. Future scientific research is needed to give us more insight into how different animals perceive certain signals or procedures. This could aid in the classification of different procedures, and additionally, these insights might help in refining certain procedures.

Throughout the tables, we aimed to give clear guidelines, with specific time periods and/or measurable clinical signs, to aid researchers, AWB and AEC members in the severity classification of animal procedures. However, the severity classification only applies to the procedure being performed once. Therefore, re-evaluation is necessary when different procedures are being combined or repeated, and a cumulative severity score needs to be given. We chose to not incorporate cumulative scores, as there are too many variables within an experimental set-up that can influence severity, e.g., frequency of and time interval between procedures. And, although repeating procedures or performing different procedures does not automatically increase severity, it might become more severe. This must, therefore, carefully be considered in order to correctly assess the severity.

Our severity classification of disease models is mostly based on clinical signs. This implies a good knowledge of the anatomy, physiology and normal behavior of the species involved. Especially the recognition of more subtle clinical signs, such as paresis and pain, might be challenging, especially as researchers in biomedical sciences may come from varying backgrounds. They thus need proper education and guidance. This applies to all species and might even be more challenging in aquatic species, such as xenopus and zebrafish, especially in their early developmental stages. We therefore included clinical signs of adult zebrafish and reported for the first-time signs to assess the early developmental stages of zebrafish (up to 12 days post fertilization). Signs to assess zebrafish between 12 days and sexual maturity and xenopus still need to be developed.

Although the aim of the EU Directive is to provide guidance, uniformity and clarity for animals involved in research, some articles of the Directive, such as the classification of procedures performed on fetuses and premature animals, remain controversial and difficult to interpret. According to the Directive, from a certain age pre-birth, a mammal fetus or larval form is considered an experimental animal as there is evidence it might experience pain. However, according to Commission Implementing Decision (EU) 2020/569 of 16 April 2020, the fetal and embryonic forms of mammalian species shall be excluded from the provision of annual statistical data. This seems in contrast with the definition of an experimental animal for which annual statistics should be provided by each EU member state. The severity classification of these early forms is therefore performed to the best of our knowledge. Similarly, some procedures in animals are not standardized in the community, making the severity classification difficult. For instance, the insulin tolerance test in mice requires a period of fasting. However, different fasting times have been reported [32]. Our severity classification takes both fasting and the influence of glycemic levels into consideration. Both these factors are affected by fasting time and, as such, might/should be classified differently. Not only for the severity estimation but also for the reproducibility of in vivo experiments, an important concern in biomedical research [33,34,35,36], further standardization of procedures is mandatory.

As new disease models, tests and GA models are developed on a daily basis, it would be interesting if more emphasis is put on how to perform a good retrospective analysis and assess actual severity. For new models or drugs, it is sometimes hard to classify prospective severity based on the available knowledge. Clear communication on the retrospective severity and, more importantly, the methods used to assess the severity level would be very informative for the research community.

As discussed above, severity classification remains a difficult task for the researchers and competent authorities during project application and evaluation, respectively. Although difficult, both prospective and actual or retrospective severity classification and reporting are important. Prospective severity is crucial in performing correct harm–benefit analysis during the project evaluation and in refining procedures. As actual severity is reported, it promotes transparency and dictates public opinion. In Belgium, project evaluation has been delegated from the competent authorities to the institutional AEC. As shown in Table 1, severity levels reported within the different countries significantly differ, and although not reported, this difference may also exist between institutions. The aim of this work was twofold. On the one hand, we wanted to align severity between two different institutions. On the other hand, we wanted to report the severity of all procedures and models used at our institutions, many of which were not covered in the available guidelines.

## 5. Conclusions

The severity classification of animal procedures remains a challenging task as it is often mostly subjective. The large variety of procedures in many different species makes it even more challenging. This is reflected in the reported number of procedures within a specific severity classification by the different EU member states, with large variations between the member states. Further evidence-based severity classification of animal procedures is needed. In the meantime, sharing experience in the field will help in further aligning the severity classification of animal procedures. This manuscript summarizes the severity classification of two Belgian academic institutions wherein animals are mainly used in biomedical research. We have provided a severity classification for all procedures and models used at our institutions and discussed the process of that classification. Many of those procedures were not covered in the available guidelines, and we hereby thus report severity classification for many procedures for the first time.

## Figures and Tables

**Figure 1 animals-13-02581-f001:**
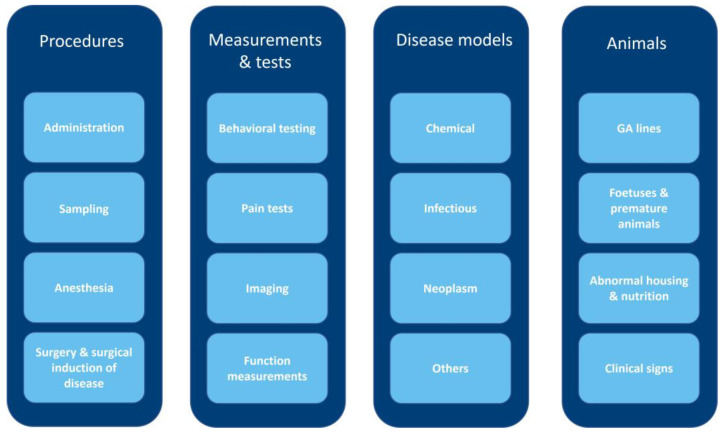
Diagram of main categories (dark blue) and subcategories (light blue)—animal procedures are subdivided into 16 subcategories. Based on commonalities, these subcategories were assigned to four main categories of procedures.

**Table 1 animals-13-02581-t001:** Severity classifications reported for the EU; Germany, France and the UK (the three countries using the most animals in Europe) and Belgium in 2020.

	Mild (%)	Moderate (%)	Severe (%)	Non-Recovery (%)	Total Number of Animals Used	Reference
EU	48.7	37.3	9.9	4.1	8,054,930	[10]
Germany	66.9	23.7	3.8	5.6	1,850,443	
France	30.1	49.8	13.7	6.4	1,643,787	
UK	51	24	4	7	1,681,383	
Belgium	55.23	31.851	10.76	2.16	437,275	

**Table 2 animals-13-02581-t002:** Involvement of the different experts in developing the severity classification system.

		Catalogue of Procedures	Extensive Review of Existing Literature and Guidelines	Decide on Approach to Severity Classification	Drawing First Draft	Reviewing First Draft Classification per Category	Drawing Final Draft per Category	Reviewing Final Draft
KU Leuven	Designated veterinarian	x	x	x	x	x	x	
	Subgroup of AEC			x		x		
	AEC							x
	Research experts				x			
VUB	Designated veterinarian	x			x	x	x	
	AWB	x				x		x
	AEC							x
	Research experts					x		

## Data Availability

Not applicable.

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
