# Peer review of "Severity Classification of Laboratory Animal Procedures in Two Belgian Academic Institutions"

_animals, 2023, doi:10.3390/ani13162581_

Round 1

Reviewer 1 Report (Previous Reviewer 2)

Thank you for your revised manuscript. I can now see how the study was conducted and you have clearly described membership of the review panel, and who was involved for specific areas and at each stage.

I still have concerns about the lack of voluntary informed consent from panel members, who may not have been aware that the findings would be published, especially as the members of the panel could potentially be identified.

However, I will leave the decision about the lack of consent to the publisher as it will fall within their requirements for publication.

The manuscript is very clearly written and uses a good style of English. Very few minor edits required.

Author Response

Thank you. I agree with leaving the decision on the consent issue to the publisher. 

Reviewer 2 Report (Previous Reviewer 1)

Congratulations to a very good paper! The revision has largely addressed my concerns with the original version, and I only have two minor comments:

1) on line 179 "has his own AWB" should be "has its own AWB"

2) I suggest that you include a reference to some of the standard papers on harm-benefit analysis (for example, the FELASA/AALAS working group papers - https://felasa.eu/working-groups/reports/id/40) at least the first time the concept is mentioned.

Author Response

1) on line 179 "has his own AWB" should be "has its own AWB"

We have adapted this in the text 

2) I suggest that you include a reference to some of the standard papers on harm-benefit analysis (for example, the FELASA/AALAS working group papers - https://felasa.eu/working-groups/reports/id/40) at least the first time the (see attachment) concept is mentioned.

Thanks for the suggestion, we have included this reference in line 66-67

This manuscript is a resubmission of an earlier submission. The following is a list of the peer review reports and author responses from that submission.

Round 1

Reviewer 1 Report

This paper presents a proposal for severity classification of a number of procedures in laboratory animals, based on work developed at two Belgian academic institutions. The paper provides a novel contribution with practical relevance. As the authors point out, the lack of standards probably leads to wide variation in classification, and detailed criteria as presented in this paper are important. The paper is clear and well written. Nevertheless, the paper would benefit from addressing a few issues as outlined in the following.

Main issues

The main limitation of the paper in its present format is that there is no information about the process behind defining the criteria. Who were the experts providing information? In which way was the information obtained? What was required for the information to be considered reliable? How much experience with a research model was considered necessary to provide expert information on severity of the model? Was there any cross-validation of information by other experts? This is important to assess the reliability of the criteria presented in the paper, and it is also helpful as a guide for others who may want to do something similar in their institutions. We have recently published a paper reporting the process of developing competency assessment scales for laboratory animal science, which I suggest you look at for an example of how to describe the process (Costa et al 2021 https://doi.org/10.1177/00236772211017767).

In the introduction and discussion, I miss a reflection on the role of the severity classification in a wider context. This reflection is important to understand why a common standard for classification is pertinent. I’m thinking of how it plays a role in decision-making over animal experiments (harm-benefit analysis) and how it indirectly contributes to public perception of animal experiments through the kind of statistics that are presented in Table 1.

Section 3.9 needs to refer to the recent Framework for the Genetically Altered Animals under Directive 2010/63/EU https://op.europa.eu/en/publication-detail/-/publication/7ff424e1-eb8f-11ec-a534-01aa75ed71a1/language-en/format-PDF/source-282223967. This document does indeed provide more detail about what is to be considered a harmful phenotype.

Minor issues

-        Terminology to describe what is usually called an animal ethics committee. On line 16 it’s referred to as “ethical commission” and on lines 37-38 “institutional ethics committee”. The standard terminology in Europe is animal ethics committee.

-        It is not true that Belgium is the only country where project evaluation takes place on the institutional level. This is also the case at least for Portugal and Italy. In Portugal the authorization is not given on the institutional level, but the important point in the context is where the project evaluation takes place.  

-        Line 150 “across” rather than “throughout”

-        Line 172 not clear which the “respective chapters” are

-        Line  190 should be “were taken into account”

-        Line 240 “and thus EU regulated” is understandable but is colloquial. A correct formal reference would be something like “and thus covered by Directive 2010/63/EU

-        Line 301-302 should rather be something like “differentiation is made between when feeding dry feed and food containing fluids with different cut-offs in time. In the latter case, water restriction is only classified as severe”

Reviewer 2 Report

Thank you for a very interesting and comprehensive paper. This was a difficult study to undertake, given the variations in EU guidelines, guidance from other countries and variations between institutional approaches to severity calculation.

You have done a very good job of explaining why you felt this study was necessary, and supported this aspect with relevant literature. The resulting tables of severity for specified procedures and situations is very helpful and should increase the accuracy of severity calculation for future projects, and for areas that are not yet included in the tables.

I feel that there are two areas where this paper needs to be improved:

1.      Methodology. The description of how this study was conducted is lacking. You have described getting various stakeholders together. We just need more information on number of people involved and their roles, how were these people selected, and how was the process organised? For example, was it based on a Delphi approach, where the proposed guidelines were refined and agreed at various meetings until a consensus was reached? At the moment it is difficult to see what was done and how agreement was reached.

2.      Informed consent. I think the informed consent of the participants is essential for this study, especially as the two institutions involved are named, and the roles of participants are clearly labelled (veterinarians and members of ethics committees). I would also expect the study to have gained ethics approval at one of the two institutions named.

At the moment it sounds like an exercise that was undertaken to solve a problem, without perhaps the intention to publish, but as it is now in press, I think that the above areas need to be addressed.